# Chimpanzees (*Pan troglodytes*) in U.S. Zoos, Sanctuaries, and Research Facilities: A Survey-Based Comparison of Species-Typical Behaviors

**DOI:** 10.3390/ani13020251

**Published:** 2023-01-10

**Authors:** Andrea W. Clay, Stephen R. Ross, Susan Lambeth, Maribel Vazquez, Sarah Breaux, Rhonda Pietsch, Amy Fultz, Michael Lammey, Sarah L. Jacobson, Jaine E. Perlman, Mollie A. Bloomsmith

**Affiliations:** 1Emory National Primate Research Center, 954 Gatewood Rd., Atlanta, GA 30322, USA; 2Lester E. Fisher Center for the Study and Conservation of Apes, Lincoln Park Zoo, 2001 North Clark Street, Chicago, IL 60614, USA; 3National Center for Chimpanzee Care, Bastrop, TX 78602, USA; 4Southwest National Primate Research Center, 8715 W. Military Dr., San Antonio, TX 78227, USA; 5New Iberia Research Center, 4401 W. Admiral Doyle Dr., New Iberia, LA 70560, USA; 6Center for Great Apes, P.O. Box 488, Wauchula, FL 33873, USA; 7Chimp Haven, 13600 Chimpanzee Pl, Keithville, LA 71047, USA; 8Coulston Foundation, Alamogordo, NM 88310, USA; 9The Graduate Center, Department of Psychology, City University of New York, 365 Fifth Avenue, New York, NY 10016, USA

**Keywords:** primate, chimpanzee, welfare, species-typical behavior, ape

## Abstract

**Simple Summary:**

A survey was sent to zoos, research facilities, and sanctuaries which housed chimpanzees. The behavioral profiles of 1122 chimpanzees were collected for this survey. Data collected included information about the animals’ age, sex, social group size, rearing history, and enclosure as well as information about each animal’s behavior. Each respondent was asked to indicate if certain behaviors had been observed in each chimpanzee over the prior two years. Species typical behaviors (STBs) were queried, including copulation, tool-use, nest-building, and social grooming. Tool-use was reported to be present for 94.3% of the sample, active grooming for 85.7%, copulation for 68.3% and nest-building for 58.9%. Male chimpanzees who were not reared by their conspecific mother were most likely to have deficits in STBs, and female chimpanzees who were mother-reared were generally the most likely to engage in STBs.

**Abstract:**

A survey was sent to zoos, research facilities, and sanctuaries which housed chimpanzees. Data collected included information about 1122 chimpanzees’ age, sex, social group-size, rearing history, and enclosure. Respondents were also asked to indicate if certain behaviors had been observed in each chimpanzee over the prior two years. Species- typical behaviors (STBs) were queried, including copulation, tool-use, nest-building, and social grooming. Tool-use was reported present for 94.3% of the sample (*n* = 982), active social grooming for 85.7% (*n* = 1121), copulation for 68.3% (*n* = 863) and nest-building for 58.9% (*n* = 982). Of the subjects for whom we had data regarding all four STBs (*n* = 860), 45.6% were reported to engage in all four. Logistic regression analyses using forward Wald criteria were conducted to determine the best model for each STB based on the predictors of age, sex, rearing history, group-size, facility-type, and a sex-by-rearing interaction. The best model for copulation (χ^2^(6) = 124.62, *p* < 0.001) included rearing, group-size, facility-type, and the sex-by-rearing interaction. Chimpanzees were more likely to copulate if they were mother-reared, in larger groups, living in research facilities, and, if not mother-reared (NOTMR), more likely to copulate if they were female. The best model for tool-use retained the predictors of age category, facility-type, and sex-by-rearing (χ^2^(5) = 55.78, *p* < 0.001). Chimpanzees were more likely to use tools if they were adult, living in research facilities, and if NOTMR, were female. The best model for nest-building included facility-type and rearing (χ^2^(3) = 205.71, *p* < 0.001). Chimpanzees were more likely to build nests if they were MR and if they were living in zoos or in sanctuaries. The best model for active social grooming retained the predictors of age, sex, rearing, and type of facility (χ^2^(6) = 102.15, *p* < 0.001). Chimpanzees were more likely to engage in active social grooming if they were immature, female, mother-reared, and living in zoos. This research provides a basic behavioral profile for many chimpanzees living under human care in the United States and allows us to determine potential methods for improving the welfare of these and future chimpanzees in this population.

## 1. Introduction

As of 2021, it is estimated that over 1300 chimpanzees are living under human care in the United States. This large population is divided into approximately 675 individuals living in sanctuaries accredited by the Global Federation of Animal Sanctuaries (GFAS), 250 animals living in zoos accredited by the Association of Zoos and Aquariums (AZA), 300 in research facilities, 150 in unaccredited facilities (both zoos and sanctuaries), and 35 living as pets, breeders or as part of the entertainment industry [1]. In the past ten years, hundreds of chimpanzees have been relocated to sanctuaries and zoos, from research facilities, private owners, the entertainment industry, and unaccredited facilities. With this diversity of housing sites, it becomes increasingly pertinent that best practices and standards of care for chimpanzees, regardless of their location, are supported by relevant and validated data. A starting point for discussion should include an assessment of the current behavioral welfare of the captive chimpanzee population.

Kagan et al. (2015) [2] proposed a universal framework for animal welfare assessments that promoted the inclusion of both input (what is provided to the animals) and output (the animals’ response). If we consider this framework in reference to chimpanzees, relevant inputs include information about the environment in which the chimpanzees are living, their ability to go indoors and outdoors, the substrate of their housing enclosures, the size of their social group and characteristics of their management, such as the use of positive reinforcement training. Ross (2020) [3] provides a comprehensive review of such input characteristics and discusses how they may impact the welfare of captive chimpanzees. Outputs are comprised of the chimpanzees’ responses to these inputs, including physiological measures and behavioral expressions. These outputs may serve as proxy for affective states, which are most easily understood as underlying, persistent emotional states [4]. Outputs are often difficult to measure and can be highly variable between individuals but are nonetheless essential to understanding the impact of our attempts to improve chimpanzee welfare.

There have been several initiatives to describe the behavioral profiles of chimpanzees with a focus on their welfare, though most of those tend to concentrate on behaviors that are indicative of negative affective states [5,6]. While welfare has often been established through an absence of atypical behavior, it is increasingly of interest to establish positive behavioral markers of welfare along with, or in lieu of, an absence of negative markers [7,8,9,10,11,12,13]. Indeed, the presence of atypical behavior, while historically considered an indicator of poor welfare [14], may not be a reliably accurate indicator of an animal’s current wellbeing for several reasons. Atypical behaviors may develop in one environment and persist in another, regardless of improvements to the environment, and usually do not develop in all animals exposed to a particular stressor [15]. Some animals may be more likely than others to develop an atypical behavior due to their individual history (i.e., early rearing experience; see i.e., early rearing experience; see [16]) regardless of the animal’s current environment. Additionally, one atypical behavior, coprophagy, has been convincingly argued to be, while undesirable, not a valid indicator of negative wellbeing [17]. However, negative indicators of welfare appear to be more salient to observers. In one study, surveys designed to measure personality and well-being, as rated by observers, were analyzed along with behavioral data collected from the same group of chimpanzees. Interestingly, while results indicated a significant association of negative indicators such as urophagy, coprophagy, and regurgitation/reingestion with an overall welfare factor, there was no significant relationship for what might be assumed to be positive indicators, such as grooming and play behavior [18]. While investigating chimpanzee atypical behavior is still important [19,20,21], species-typical behaviors (STBs) such as grooming and playing with other chimpanzees should be included in future studies.

Species-typical behaviors are observed in a species’ wild-living counterparts and are often evaluated as markers of positive welfare [10,12,13,22]. However, these indicators, like atypical behaviors, are not infallible. For example, while copulation and maternal care are considered STBs, Cronin et al. (2020) [23] found no link between a chimpanzee’s opportunity to reproduce and the expression of atypical behavior. The authors assert that the absence of such evidence does not necessarily mean that chimpanzees do not experience behavioral benefits from breeding and rearing young and emphasize that it is difficult to ascertain how much behavior, or which behavior, would indicate positive welfare. In this vein, some previous studies have noted several behavioral similarities between wild and captive chimpanzee groups [24,25,26], which does not necessarily indicate similar welfare states for wild and captive chimpanzees. Another study noted fewer negative impacts on social behavior than were expected based on captive chimpanzees’ rearing history [27], which does not necessarily indicate that mother-reared and non-mother-reared chimpanzees experience equivalent welfare.

Both atypical and species-typical behaviors may be impacted by an animal’s history, but the effect of that history on current behavior may depend on the specific behavior being observed. Some STBs, such as copulation, may be resistant to change, regardless of the environment [16]. This is analogous, then, to rocking, an atypical behavior often noted to persist beyond the environment in which it originated [20,28]. Clearly, both atypical and species-typical behaviors have limitations as indicators of an animal’s current welfare. There are many complex issues that inform which behaviors may be positive, negative, or merely different from what may be observed for a species in its natural environment [8]. It is not a simple matter to evaluate welfare of a species in the wild, much less to use similar evaluative strategies to assess animals in captive conditions [29]. Behavioral flexibility [8,9,30] and behavioral diversity [31] are two measures that show promise as welfare assessment procedures, as does the study of affective states and methods for measuring them. Research should continue to focus on an understanding of animal welfare beyond the behaviors we currently use as proxy for affective states (e.g., [4]. Currently, however, our most accessible option for assessing welfare still relies on observed behavior, despite the limitations outlined here.

Toward that end, we conducted a survey designed to gather basic information about the environment and behavior of over one thousand chimpanzees living in the United States, focusing on easily observable behaviors which can be operationalized and recognized by observers familiar and unfamiliar with chimpanzee behavior. As part of this survey, we asked about four species-typical behaviors. The STBs queried were copulation, tool-use, nest-building, and active social grooming.

Copulatory behavior is of primary biological importance for all animals and as such, has presumed associations with welfare in part because such behavior in chimpanzees is demonstrably influenced by early social environments [16,25]. Tool-use is a complex ability seen in chimpanzees in both natural [32,33] and captive [34,35,36] settings and has been labeled as a marker of behavioral flexibility. Examples of wild chimpanzees engaging in such varied, flexible behavior (i.e., tool-use) include the use of rocks to crack open nuts, twigs or grass to obtain termites from termite mounds, and crushed leaves to absorb water for drinking [37]. Nest-building is highly typical in adult wild chimpanzees [38] who express this behavior virtually every day [39] and while it is also observed in most captive living chimpanzees [38,40], it has been characterized as less complex for chimpanzees not reared with conspecifics [41,42]. Social grooming is an essential part of chimpanzee social life [43] due to its role in the creation and maintenance of social bonds [44,45]. Of note, we asked about the chimpanzees’ active social grooming as an indicator of their wellbeing, rather than simply the chimpanzees’ reception of grooming by a social partner, as the active grooming of a social partner can function as part of an exchange for social support [46] or social tolerance [47].

While there are myriad other STBs which we may have selected, we chose these four behaviors as they occur frequently in wild chimpanzees, may be linked to positive welfare, and are easily identified by observers. Social play and aggressive behavior, for example, are also behaviors observed frequently in wild chimpanzees, and may be arguably more important for an animal’s welfare. However, these behaviors are often nuanced and/or subtle and may not be as consistently and easily identified by observers. We expected to collect information from a variety of observers, not all of whom would have the same understanding of chimpanzee behavior.

We collected information about chimpanzees living at multiple facilities, including accredited zoos, sanctuaries, and research facilities. Information gathered from so many different facilities is potentially helpful in identifying areas in which progressive management systems can improve. Investigating the factors which may be associated with the recent expression of STBs cannot distinguish between the proximate and historical influences of that behavior. We may not be able to discern whether a behavior occurs at low frequencies because of a chimpanzee’s current circumstances or, rather, due to his or her past experiences. However, our aim here is to characterize the current prevalence of four species-typical behaviors in chimpanzees, investigate possible factors that may affect their expression, identify areas for future research, and ultimately promote environments and animal management that will support the expression of these behaviors moving forward.

## 2. Materials and Methods

As part of a larger, survey-based study (see Appendix A for complete survey) of chimpanzee behavior, questionnaires were sent to twenty-six AZA-accredited zoological parks, two GFAS-accredited sanctuaries, and seven research facilities in the United States which housed chimpanzees in 2015. This covered all United States research facilities and AZA-accredited zoos that held chimpanzees at the time as well as two GFAS-accredited sanctuaries; one other sanctuary was approached but they did not have available staff to complete the survey. We asked respondents about each individual chimpanzee living at that facility: enclosure characteristics, social group size, rearing history, age, and sex (summarized elsewhere [19,20]). Group-size was queried because the complexity of a chimpanzee social group has been determined to be relevant to chimpanzee welfare in captivity [48], supported by reports of group-size impact on social and object-directed behaviors [49]. Group-size was reported at the time the survey was completed by each facility. We did not ask for information regarding changes in group-size, if they occurred during the two-year period, as that would have been prohibitively labor intensive for those completing the surveys (who, in some cases, compiled information for more than 100 individual chimpanzees at their facility). All respondents were asked to answer, yes or no, if the specific STBs (copulation, tool-use, nest-building, and active social grooming) had been observed at least once in the past two-year period. We chose a two-year period of assessment to increase the likelihood that some of the more infrequent behaviors might be observed by staff members without extending the period of assessment such that retrieving records would be difficult for those completing the surveys. The survey was sent to an individual at each institution, and the surveys were completed as each institution saw fit. There were many possible sources of information that could be used, and the institutions were asked to report the source(s) they used from the following list of options: quantitative behavioral data, routine behavioral monitoring information, records such as animal training records or other archived information, observations by behavioral staff or others, and/or other sources (Appendix A).

It is important to note that some proportion of subjects had spent significant portions of their lives in different facilities (i.e., most of the subjects in sanctuaries had lived in research facilities, and some subjects in zoos and sanctuaries moved to their current location after living as a pet or were involved in entertainment). We did not collect data as to subjects’ origin other than pertaining directly to rearing history in the first year of life, so did not include a separate origin factor as a potential predictor, though at least one study has found origin, in addition to rearing history, to be relevant to adult chimpanzee atypical behaviors as observed in zoos [21]. Nonetheless, facility-type was included in regression analyses because, while there is a great deal of variation within each type of facility, there are also some commonalities, e.g., all zoos will have visitors, and all sanctuaries will be populated by animals who almost exclusively have been translocated from private ownership or research facilities.

We did not attempt to evaluate observer reliability. Information was compiled from many potential people and sources at any one institution. We did not query about frequency of behavior and chose relatively easy-to-identify behaviors to control for the facilities’ different methods of observation and documentation of chimpanzee behavior. Surveys were distributed and returned over a period of about six months.

### Analysis

Subjects’ ages were categorized as immature (2–11 years old), adult (12–39 years old), and elderly (40 years or older). Respondents provided the age of each chimpanzee at the time the survey was filled out, so it was possible that a subject may be categorized as an adult at 12 years of age even though some of the years during which their behavior was assessed would have fallen in the ‘immature’ age category. Subjects were considered mother-reared (MR) if they had spent at least half of their first year of life with a conspecific mother [16]. Wild-born subjects were categorized as mother-reared as it was generally true that infants taken from the wild at less than six months of age would not have survived. Subjects for whom rearing history was known, but who did not fit the definition for mother-reared, were scored as not mother-reared (NOTMR). Subjects whose rearing history was not known were scored as unknown (UNK) and removed from regression analyses which included rearing as a possible predictor. These subjects were retained for chi-square tests so that comparisons based on age, sex, group-size, and facility would include them, but any significant differences pertaining specifically to UNK rearing will not be further discussed here (they are included in results tables). Subjects’ group-size was categorized into bigger groups (8 or more chimpanzees), smaller groups (3–7 chimpanzees) and pairs [50]. One chimpanzee had been very recently singly housed but was included in the pair group as he was paired for the two years covered by the survey.

Information regarding the survey subjects’ age, sex, and rearing history are summarized in Appendix A. Demographics by facility type are reported in Appendix A, and subjects’ rearing history and sex by group-size are reported in Appendix A. Mean age and subjects’ group size by facility type are reported in Appendix A (Appendix A, Appendix A, Appendix A and Appendix A are included in Appendix B). We calculated the overall percent of the surveyed population (*n* = 1122) reported to engage in each STB. Note that for these STBs there were some subjects removed from each calculation. Some facilities did not respond regarding all behaviors, for example, if the person filling out the survey had not seen the chimpanzees with tools available or did not provide nest-building material. We also removed 15 subjects who were less than five years old or living, for the past two years, without access to the opposite sex (we did not query if available opposite sex groupmates were related to the subject), from the total subject pool for copulation. Sample sizes in each assessed category with data pertaining to each STB are included in Table 1 (for regression analyses) and Table 2, Table 3, Table 4, Table 5 and Table 6 (for chi-square tests) in the results section of this paper.

We assessed the relationship between the possible predictors of age, sex, rearing history, facility-type and current (at the time the survey was filled out) social group-size with the presence or absence of four STBs: (1) copulation; (2) tool-use; (3) nest-building and (4) active social grooming. For each STB, we conducted a logistic regression using forward-Wald criteria to determine the most complex model which was (1) significant at the step adding the last predictor, (2) significant as a model, and (3) passed the Hosmer and Lemeshow (HL) goodness-of-fit test. All predictors tested were first assessed for collinearity [51] by entering them into a linear regression per STB (VIF < 2.5, tolerance = 1(±0.2), correlation < 0.8) and rejected if highly collinear. Using a regression in this manner is an accepted method for exploration of data [52,53] and while there are other methods for modeling, we chose to use the same procedure as had been used in previous analyses of the same body of data. This allowed us to assess the relative strengths of the relationships between age, sex, rearing history, group-size, facility-type, and an interaction between sex and rearing history (as had been relevant in previous analyses [19,20]) with each STB surveyed. Reference groups selected were female (for sex), adult (for age), MR (for rearing history), smaller group (for group-size) and research facility (for facility-type). We also conducted a logistic regression to determine the impact of the same predictors on likelihood of a subject engaging in all four STBs, considering only subjects for whom data regarding all four behaviors were reported.

We conducted chi-square analyses for each STB comparing proportions of the sample engaging in each STB based on the same categories used as predictors in the regression analyses. This allowed us to assess a larger sample, as subjects with unknown rearing history could be included. Conducting a chi-square procedure also allowed for the determination of any significant differences between non-reference categories which were not compared by regression procedures (immature versus elderly subjects, zoo versus sanctuary residents, and paired subjects versus those living in bigger groups). Additionally, MR subjects were divided into CB (captive-born) and WB (wild-born), and chi-square tests were used to compare MR/CB to MR/WB on each of the dependent STB measures as we had the unique ability to characterize a large group of wild-born individuals across several types of environments. Subjects in the immature age category were not included in these comparisons because there were no immature wild-born subjects; this resulted in the removal of 61 subjects. Bonferroni-corrected significance due to multiple comparisons between categories (at an alpha of 0.05) is reported here for pairwise comparisons post-hoc when compared categories numbered more than two. A Cramer’s V (CV) measure of association between independent variables and STBs is reported along with chi-square values. Due to the number of tests, we used exact tests of significance for all chi-square comparisons.

Building on chi-square tests comparing MR/CB to MR/WB subjects, we conducted logistic regression analyses to determine the relative strength and predictive value of age and rearing (MR/CB vs. MR/WB) on the STBs where chi-square had found significantly different probabilities based on the CB versus WB comparison. MR/CB and adult categories served as references for these regression tests.

While regression analyses were intended to determine the relative impact of several factors on STBs, chi-square tests were intended to provide information about significant or nonsignificant differences between categories (e.g., MR versus NOTMR), regardless of the relative strength of impact per factor (e.g., rearing versus age). All analyses were conducted using SPSS (v.26) data analysis software.

## 3. Results

### 3.1. Overview

A total of 35 facilities to whom we sent surveys participated in this project. One sanctuary was unable to participate due to lack of staff. Mean age of chimpanzees sampled was 26.9 (SD = 11.4, Min = 2, Max = 77). Mean group-size (based on size of the group a subject lived in) was 7.4 (SD = 4.1, Min = 1, Max = 23). Tool-use was reported present for 94.3% of the sample (*n* = 982), active social grooming for 85.7% (*n* = 1121), copulation for 68.3% (*n* = 863) and nest-building for 58.9% (*n* = 982). Of the subjects for whom we had data regarding all four STBs (*n* = 860), 45.6% were reported to engage in all four.

Details regarding each regression analysis’ best model statistical values are presented in Table 1. All predictors were first entered into linear regression analyses to check for collinearity issues and retained for logistic regression only if the tolerance and VIF numbers were appropriate [54]. No predictors were removed due to collinearity problems. Strength and direction of the predictors’ effect on the occurrence of each STB, when retained by the best model and significant, are summarized in Table 7. Chi-square statistics and sample sizes are summarized in Table 2, Table 3, Table 4, Table 5 and Table 6.

**Table 2 animals-13-00251-t002:** Chi-square statistics for copulation.

Independent Variable	Categories Tested	Dependent Variable and Statistics						
		Copulation						
		# Y	*n*	% Y	Χ^2^	df	*p*	CV
Age	Immature	35	61	57.4	4.81	2	0.090	0.075
	Adult	477	683	69.8				
	Elderly	77	119	64.7				
Sex	Female	369	504	**73.2 ^a^**	13.78	1	**0.000**	0.126
	Male	220	359	**61.3 ^b^**				
Rearing	MR	355	434	**81.8 ^a^**	79.86	2	**0.000**	0.304
	NOTMR	202	354	**57.1 ^b^**				
	UNK	32	75	**42.7 ^b^**				
Within MR	Female	213	258	82.6	0.25	1	0.704	0.024
	Male	142	176	80.7				
Within NOTMR	Female	133	200	**66.5 ^a^**	16.71	1	**0.000**	0.217
	Male	69	154	**44.8 ^b^**				
Within UNK	Female	23	46	50.0	2.62	1	0.151	0.187
	Male	9	29	31.0				
WB vs. CB	MR/CB	266	304	**87.5 ^a^**	11.00	1	**0.001**	0.168
	MR/WB	61	84	**72.6 ^b^**				
Group-size	Pair	37	62	59.7	6.30	2	**0.042**	0.085
	Smaller	306	463	66.1				
	Bigger	92	246	72.8				
Facility-type	Zoo	123	195	**63.1 ^a^**	42.95	2	**0.000**	0.223
	Research	369	480	**76.9 ^b^**				
	Sanctuary	97	188	**51.6 ^a^**				

# Y = # subjects reported to engage in the behavior; *n* = total subjects for which data existed; % Y = % of *n* (total) subjects reported to engage in behavior; X^2^ = chi-square value; df = degrees freedom; *p* = probability/chi-square; CV = Cramer’s V measure of association between independent and dependent variable. Bolded text indicates significant differences between categories based on a chi-square test. Superscripts indicate column proportions significantly different pairwise after Bonferroni correction at alpha = 0.05.

**Table 3 animals-13-00251-t003:** Chi-square statistics for tool-use.

Independent Variable	Categories Tested	Dependent Variable and Statistics						
		Tool-Use						
		# Y	*n*	% Y	Χ^2^	df	*p*	CV
Age	Immature	92	112	**82.1 ^a^**	34.74	2	**0.000**	0.188
	Adult	714	745	**95.8 ^b^**				
	Elderly	120	125	**96.0 ^b^**				
Sex	Female	524	546	**96.0 ^a^**	6.40	1	**0.013**	0.081
	Male	402	436	**92.2 ^b^**				
Rearing	MR	452	469	**96.4 ^a^**	11.31	2	**0.003**	0.107
	NOTMR	384	411	**93.4 ^a,b^**				
	UNK	90	102	**88.2 ^b^**				
Within MR	Female	261	270	96.7	0.16	1	0.804	0.018
	Male	191	199	96.0				
Within NOTMR	Female	218	229	95.2	2.63	1	0.113	0.080
	Male	166	182	91.2				
Within UNK	Female	45	47	**95.7 ^a^**	4.74	1	**0.034**	0.215
	Male	45	55	**81.8 ^b^**				
WB vs. CB	MR/CB	312	322	96.9	0.15	1	0.751	0.019
	MR/WB	84	86	97.7				
Group-size	Pair	66	74	89.2	3.90	2	0.149	0.063
	Smaller	510	538	94.8				
	Bigger	350	370	94.6				
Facility-type	Zoo	183	204	**89.7 ^a^**	17.26	2	**0.000**	0.133
	Research	528	561	**94.1 ^a^**				
	Sanctuary	215	217	**99.1 ^b^**				

# Y = # subjects reported to engage in the behavior; *n* = total subjects for which data existed; % Y = % of *n* (total) subjects reported to engage in behavior; X^2^ = chi-square value; df = degrees freedom; *p* = probability/chi-square; CV = Cramer’s V measure of association between independent and dependent variable. Bolded text indicates significant differences between categories based on a chi-square test. Superscripts indicate column proportions significantly different pairwise after Bonferroni correction at alpha = 0.05.

**Table 4 animals-13-00251-t004:** Chi-square statistics for nest-building.

Independent Variable	Categories Tested	Dependent Variable and Statistics						
		Nest-Building						
		# Y	*n*	% Y	Χ^2^	df	*p*	CV
Age	Immature	33	112	**29.5 ^a^**	81.46	2	**0.000**	0.288
	Adult	436	745	**58.5 ^b^**				
	Elderly	109	125	**87.2 ^c^**				
Sex	Female	364	546	**66.7 ^a^**	30.95	1	**0.000**	0.178
	Male	214	436	**49.1 ^b^**				
Rearing	MR	354	469	**75.5 ^a^**	106.10	2	**0.000**	0.329
	NOTMR	188	411	**45.7 ^b^**				
	UNK	36	102	**35.3 ^b^**				
Within MR	Female	220	270	**81.5 ^a^**	12.39	1	**0.001**	0.163
	Male	134	199	**67.3 ^b^**				
Within NOTMR	Female	123	229	**53.7 ^a^**	13.24	1	**0.000**	0.179
	Male	65	182	**35.7 ^b^**				
Within UNK	Female	21	47	44.7	3.36	1	0.096	0.182
	Male	15	55	27.3				
WB vs. CB	MR/CB	249	322	**77.3 ^a^**	6.30	1	**0.015**	0.124
	MR/WB	77	86	**89.5 ^b^**				
Group-size	Pair	51	74	**68.9 ^a^**	45.41	2	**0.000**	0.215
	Smaller	265	538	**49.3 ^b^**				
	Bigger	262	370	**70.8 ^a^**				
Facility-type	Zoo	188	204	**92.2 ^a^**	121.22	2	**0.000**	0.351
	Research	270	561	**48.1 ^b^**				
	Sanctuary	120	217	**55.3 ^b^**				

# Y = # subjects reported to engage in the behavior; *n* = total subjects for which data existed; % Y = % of *n* (total) subjects reported to engage in behavior; X^2^ = chi-square value; df = degrees freedom; *p* = probability/chi-square; CV = Cramer’s V measure of association between independent and dependent variable. Bolded text indicates significant differences between categories based on a chi-square test. Superscripts indicate column proportions significantly different pairwise after Bonferroni correction at alpha = 0.05.

**Table 5 animals-13-00251-t005:** Chi-square statistics for active social grooming.

Independent Variable	Categories Tested	Dependent Variable and Statistics						
		Active Social Grooming						
		# Y	*n*	% Y	Χ^2^	df	*p*	CV
Age	Immature	109	112	**97.3 ^a^**	13.95	2	**0.001**	0.112
	Adult	719	854	**84.2 ^b^**				
	Elderly	133	155	**85.8 ^b^**				
Sex	Female	566	623	**90.9 ^a^**	30.09	1	**0.000**	0.164
	Male	395	498	**79.3 ^b^**				
Rearing	MR	475	497	**95.6 ^a^**	80.77	2	**0.000**	0.268
	NOTMR	384	478	**80.3 ^b^**				
	UNK	102	146	**69.9 ^c^**				
Within MR	Female	280	285	**98.2 ^a^**	11.28	1	**0.001**	0.151
	Male	195	212	**92.0 ^b^**				
Within NOTMR	Female	224	261	**85.8 ^a^**	10.97	1	**0.001**	0.151
	Male	160	217	**73.7 ^b^**				
Within UNK	Female	62	77	**80.5 ^a^**	8.79	1	**0.004**	0.245
	Male	40	69	**58.0 ^b^**				
WB vs. CB	MR/CB	323	336	96.1	1.73	1	0.273	0.063
	MR/WB	93	100	93.0				
Group-size	Pair	59	86	**68.6 ^a^**	30.38	2	**0.000**	0.165
	Smaller	508	601	**84.5 ^b^**				
	Bigger	394	434	**90.8 ^c^**				
Facility-type	Zoo	196	204	**96.1 ^a^**	21.87	2	**0.000**	0.140
	Research	584	701	**83.3 ^b^**				
	Sanctuary	181	216	**83.8 ^b^**				

# Y = # subjects reported to engage in the behavior; *n* = total subjects for which data existed; % Y = % of *n* (total) subjects reported to engage in behavior; X^2^ = chi-square value; df = degrees freedom; *p* = probability/chi-square; CV = Cramer’s V measure of association between independent and dependent variable. Bolded text indicates significant differences between categories based on a chi-square test. Superscripts indicate column proportions significantly different pairwise after Bonferroni correction at alpha = 0.05.

**Table 6 animals-13-00251-t006:** Chi-square statistics for all four STBs.

Independent Variable	Categories Tested	Dependent Variable and Statistics						
		All Four STBs						
		# Y	*n*	% Y	Χ^2^	df	*p*	CV
Age	Immature	16	61	**26.2 ^a^**	13.89	2	**0.001**	0.127
	Adult	310	680	**45.6 ^b^**				
	Elderly	66	119	**55.5 ^b^**				
Sex	Female	262	503	**52.1 ^a^**	20.68	1	**0.000**	0.155
	Male	130	357	**36.4 ^b^**				
Rearing	MR	275	434	**63.4 ^a^**	116.88	2	**0.000**	0.369
	NOTMR	106	354	**29.9 ^b^**				
	UNK	11	72	**15.3 ^c^**				
Within MR	Female	172	258	66.7	2.99	1	0.086	0.083
	Male	103	176	58.5				
Within NOTMR	Female	79	200	**39.5 ^a^**	20.02	1	**0.000**	0.238
	Male	27	154	**17.5 ^b^**				
Within UNK	Female	11	45	**24.4 ^a^**	7.79	1	**0.005**	0.329
	Male	0	27	**0.0 ^b^**				
WB vs. CB	MR/CB	207	304	68.1	0.743	1	0.432	0.044
	MR/WB	53	84	63.1				
Group-size	Pair	29	60	**48.3 ^a,b^**	36.00	2	**0.000**	0.205
	Smaller	168	462	**36.4 ^a^**				
	Bigger	195	338	**57.7 ^b^**				
Facility-type	Zoo	102	195	**52.3 ^a^**	12.81	2	**0.002**	0.122
	Research	225	478	**47.1 ^a^**				
	Sanctuary	65	187	**34.8 ^b^**				

# Y = # subjects reported to engage in the behavior; *n* = total subjects for which data existed; % Y = % of *n* (total) subjects reported to engage in behavior; X^2^ = chi-square value; df = degrees freedom; *p* = probability/chi-square; CV = Cramer’s V measure of association between independent and dependent variable. Bolded text indicates significant differences between categories based on a chi-square test. Superscripts indicate column proportions significantly different pairwise after Bonferroni correction at alpha = 0.05.

**Table 7 animals-13-00251-t007:** Strength and direction of impact of predictors per STB based on forward-Wald logistic regression Exp(B)-values for significant (*p* < 0.05) predictors.

Predictor:	Age (Compared to Adults)		Sex (Compared to Females)	Rearing (Compared to Mother-Reared)	Facility-Type (Compared to Research Facility)		Group-Size (Compared to Smaller Group)		Sex by Rearing (Compared to MR/Females)
STB:	Immature	Elderly	Male	Not MR	Zoo	Sanctuary	Pairs	Bigger Group	Not MR, Male
Copulation	--	--	--	0.46x	0.29x	0.27x	--	**2.13x**	0.38x
Tool-use	0.12x	--	--	--	0.34x	--	--	--	0.27x
Nest-building	--	--	--	0.27x	**13.7x**	**2.0x**	--	--	--
Active Social Groom	**5.3x**	--	0.41x	0.18x	**3.1x**	--	--	--	--
All Four STBs	0.21x	--	--	0.32x	--	--	--	**2.85x**	0.33x

Bolded values indicate higher likelihood compared to reference category; values not bolded indicate a lower likelihood compared to reference category; x = chance of individual in this category engaging in the associated behavior as compared to the reference category, e.g., individuals who were immature were 0.12 times as likely to engage in tool use as individuals who were adult but were 5.3 times as likely as adults to engage in active social grooming.

### 3.2. Copulation

The best model for copulation retained rearing, group-size, facility-type, and sex-by-rearing (Table 1) as significant predictors (χ^2^(6) = 124.62, *p* < 0.001; −2 log likelihood = 828.77; CS (Cox-Snell) R^2^ = 0.15, N (Nagelkerke) R^2^ = 0.21; HL: χ^2^(7) = 8.09, *p* = 0.324). While the regression continued to add significant predictors beyond that point, doing so caused the model to fail the HL test, so those models are not reported. NOTMR subjects were 0.46 times as likely as MR to copulate, and subjects living in bigger groups were 2.13 times more likely to copulate than those in smaller groups. Subjects in zoos were 0.29 times as likely as those in research facilities and those in sanctuaries were 0.27 times as likely as those in research facilities to copulate. NOTMR subjects who were also male were 0.38 times as likely as MR females to copulate (Table 7).

Chi-square tests found significant differences in likelihood of copulation based on an individual’s sex, rearing history, group-size, and facility-type. Females were more likely to copulate than males, MR more likely than NOTMR, and chimpanzees living in research facilities were more likely than those in zoos or sanctuaries to copulate. Within NOTMR rearing category, females were more likely to copulate than males. Within the category of MR, MR/CB chimpanzees were more likely to copulate than MR/WB (Table 2).

### 3.3. Tool-Use

The best model for tool-use (Table 1) retained the predictors of age category, facility-type, and sex-by-rearing (χ^2^(5) = 55.78, *p* < 0.001; −2 log likelihood = 293.61; CS R^2^ = 0.06, N R^2^ = 0.19; HL: χ^2^(5) = 7.36, *p* = 0.195). Immature subjects were 0.12 times as likely to use tools as adults. Subjects in zoos were 0.34 times as likely as those in research facilities to use tools. NOTMR male subjects were 0.27 times as likely to use tools as MR female subjects (Table 7). Chi-square results indicated that immature subjects were significantly less likely to be reported as using tools as compared to adults and as compared to elderly subjects. Females were significantly more likely to use tools than males. Sanctuary-living chimpanzees were significantly more likely to use tools than either of the other categories (Table 3). Considered together, analyses indicate that the sex-by-rearing factor had the most impact on tool-use and that age had consistent impact, as well. Facility-type had significant impact, but the strength of that impact and difference between each type of facility is likely complicated by other factors such as proportion of animals at each facility who are elderly, or who were NOTMR.

### 3.4. Nest-Building

The best model for nest-building (Table 1) retained only facility-type, but the next step, which added rearing, was reasonably close to passing the HL test and accounted for up to 10% more variance in the dependent measure, so we report that model here (χ^2^(3) = 205.71, *p* < 0.001; −2 log likelihood = 966.51; CS R^2^ = 0.21, N R^2^ = 0.28; HL: χ^2^(4) = 10.83, *p* = 0.029). While the regression continued to find significant predictors to add beyond rearing, doing so caused the model to fail the HL test at p-values below 0.005. NOTMR subjects were 0.27 times as likely as MR to build nests. Subjects in zoos were 13.7 times more likely than those in research facilities, and subjects in sanctuaries were 2.0 times more likely than those in research facilities to build nests (Table 7).

Chi-square tests detected a significant difference in nest-building (*n* = 982) based on age, sex, rearing, group-size, facility-type, and sex-by-rearing category. Elderly chimpanzees were significantly more likely to build nests than adults or immatures, and adults were more likely than immatures. Females were found significantly more likely to build nests than males. MR subjects were significantly more likely to build nests than NOTMR. Chimpanzees in smaller groups were significantly less likely to build nests than those in pairs or bigger groups. Zoo-living subjects were significantly more likely to build nests than subjects living in research facilities or in sanctuaries. Within the rearing categories of MR and NOTMR, females were more likely to build nests than males. Within the MR category, MR/WB chimpanzees were significantly more likely to build nests than MR/CB (Table 4).

Considered together, analyses indicate that facility type had the strongest impact on nest-building, followed by rearing. Comparisons which did not include facility type revealed other important differences based on age, sex, group size, and sex-by-rearing. As we did not ask about the provision of nesting material, aside from requesting that respondents did not answer questions about behaviors which were not possible to observe (e.g., if no tools were provided or no nesting materials available), we cannot be sure what is driving the observed difference in nesting behavior based on facility type.

### 3.5. Social Grooming

The best model for active social grooming (Table 1) retained the predictors of age, sex, rearing, and type of facility (χ^2^(6) = 102.15, *p* < 0.001; −2 log likelihood = 609.36; CS R^2^ = 0.10, N R^2^ = 0.20; HL: χ^2^(7) = 9.80, *p* = 0.200). Immature chimpanzees were 5.3 times more likely than adults to engage in active social grooming, males were 0.41 times as likely as females, NOTMR were 0.18 times as likely as MR, and subjects in zoos were 3.1 times more likely than those in research facilities to actively groom (Table 7).

Based on a chi-square, immature chimpanzees were more likely to engage in active social grooming than adults or elderly animals and females were significantly more likely than males. MR animals were significantly more likely to engage in active social grooming than NOTMR. Subjects in bigger groups were more likely to actively groom than those in smaller groups or pairs, and subjects in smaller groups were more likely than those in pairs. Zoo subjects were significantly more likely to actively groom than research subjects or sanctuary subjects. Within rearing categories (MR, NOTMR), females were more likely to actively groom than males (Table 5).

Here, both analyses confirm significant impact of age, sex, rearing, and facility-type. Age and rearing appear to have the strongest impact on active social grooming in this case, but sex has a consistently significant impact even if other factors are accounted for. The consistent finding that zoo subjects were more likely to groom than subjects living at other facilities, even when other factors were accounted for, suggests that some unqueried factor is important here. This could be related to the nature of chimpanzee social groupings at zoos (such as male: female ratios, which we did not ask about), for example.

### 3.6. All Four STBs

The best model for a subject engaging in all four STBs (Table 1) retained the predictors of age, rearing, group-size, and sex-by-rearing (χ^2^(6) = 174.83, *p* < 0.001; −2 log likelihood = 916.72; CS R^2^ = 0.20, N R^2^ = 0.27; HL: χ^2^(6) = 5.90, *p* = 0.434). Immatures were 0.21 times as likely as adults to exhibit all four, NOTMR were 0.32 times as likely as MR, and subjects in bigger groups were 2.85 times more likely than those in smaller groups to exhibit all four STBs. The interaction predictor indicates that chimpanzees who were NOTMR and male were 0.33 times as likely as MR females to engage in all four STBs (Table 7).

Chi-square tests also found a significant difference in a subject’s likelihood of exhibiting all four STBs based on age category: Immatures were found less likely to engage in all four STBs than either adults or elderly subjects. Females were found more likely to engage in all four STBs than males. MR subjects were more likely than NOTMR. Animals living in bigger groups were more likely than those in smaller groups. Sanctuary-living chimpanzees were less likely than those in research facilities or in zoos to engage in all four STBs. Within the rearing category of NOTMR, females were more likely than males (Table 6).

Both analyses found significant impact of age, sex, rearing, group-size, and sex-by-rearing. When these factors were accounted for, facility-type did not continue to add significant impact. When sex-by-rearing was accounted for, sex apart from rearing did not have significant impact. The strongest impact on the likelihood that an animal engaged in all four STBs was related to being NOTMR and male.

### 3.7. Mother-Reared in Captivity versus in the Wild

There was a total of 388 subjects in the MR category who were adult (*n* = 301) or elderly (*n* = 87); of those, 84 were wild born (WB) and 304 were born in captivity (CB). WB and CB subjects did not significantly differ in proportion of subjects reported to use tools, engage in active social grooming, or engage in all four STBs. However, a significantly larger proportion of CB were reported to copulate, and a significantly larger proportion of WB subjects were reported to build nests (Table 2, Table 3, Table 4, Table 5 and Table 6).

A logistic regression was conducted using a forward-Wald procedure to determine the predictive value of age (as adult or elderly, adult as the reference category) versus rearing (CB as reference compared to WB) on nest-building and copulation for subjects categorized as MR. Results indicated that while age, not rearing, was significantly predictive of copulation (B (elderly) = −1.09, SE = 0.30, Wald (1) = 13.50, *p* < 0.001; elderly 0.34 times as likely as adults to be reported as copulating at least once over the past two years), rearing was significantly predictive, and age was not, for nest-building (B (WB) = 0.920, SE = 0.38, Wald (1) = 5.96, *p* = 0.015; WB 2.51 times more likely to build nests than CB). The model for nest-building retaining rearing (WB or CB) as a predictor was significant (χ^2^(1) = 7.07, *p* = 0.008; −2 log likelihood = 402.36; CS R^2^ = 0.02, N R^2^ = 0.03; passed HL and collinearity tests) as was that for copulation retaining age (adult or elderly) (χ^2^(1) = 12.81, *p* < 0.001; −2 log likelihood = 324.77; CS R^2^ = 0.03, N R^2^ = 0.06; passed HL and collinearity tests).

## 4. Discussion

We assessed a large population of captive-living chimpanzees for the presence, or absence, of four species-typical behaviors with the use of a survey. We found that a chimpanzees’ age, sex, rearing history, current facility-type, and social group-size were relevant to the presence of STBs, and that the most consistent negative impact on STB was associated with sex (being male) and rearing history (NOTMR). The ability to analyze data from such a large sample of chimpanzees under human care in the United States may allow us to make some headway in understanding and improving chimpanzee management and promoting increased expression of these natural behaviors.

### 4.1. Copulation

Only 68.5% of the 863 chimpanzees for whom we had data were reported to copulate. A previous study reported that nursery-reared subjects engaged in less frequent sexual behavior than mother-reared, though as they aged, this difference was less notable [25]. We did not ask for information related to frequency of any of the STBs assessed by survey, so a direct comparison is not possible here. However, we did not find any significant effect of age on copulation for the larger sample assessed by this study. A smaller sample including only MR adults and elderly animals was analyzed for impact of age and WB versus CB rearing history on copulation: elderly animals were significantly less likely to copulate, but WB versus CB did not significantly impact copulation.

Considering the sex of the subject, males in a previous study by King et al. [16] were more likely to copulate: 76.0% of males were observed copulating versus 61.3% in our study. Females in the two studies were relatively equivalent in this regard (72.3% of our sample copulated versus 74% in the earlier study).

King et al. (1997) [16] also observed that 92.7% of mother-reared chimpanzees, defined as with their mother for at least 12 months, copulated, compared to 81.1% of our sample. The definitions of ‘mother-reared’ and ‘non-mother-reared’ varied across the two studies, however. Our NOTMR (57.1% copulated) group included all chimpanzees who were not with their mothers for at least 6 of the first 12 months of life, while King et al. (1997) divided non-mother-reared animals into those removed at less than 1 month of age (44.4% copulated) and those removed between 1 and 12 months (58.3% copulated) [16]. These methodological differences may explain the difference in findings between the two studies as subjects we categorized as MR would have been potentially categorized as non-mother-reared by King et al. (1997) [16].

Chimpanzees with less chimpanzee interaction and more human interaction during early life, such as are presumed present for most NOTMR subjects, have been found to exhibit lower levels of sexual behavior as adults [55]. Our finding that NOTMR subjects were less likely to copulate than MR corroborates this. However, it is not clear why non-mother-rearing would differentially impact males and females in this regard. We found a sex difference within the NOTMR category (66.5% of NOTMR females copulated versus 44.8% of NOTMR males), but this was not the case within the MR category. This sex difference deserves more attention.

Our findings indicate that animals living in larger groups (8 or more individuals) are more likely to copulate (72.8%) than animals in small groups (3 to 7 chimpanzees: 66.1%). This may be due to the availability of more potential sexual partners and/or the selection (by staff) of more socially capable chimpanzees for larger groups. A smaller percentage of animals in pairs (59.7%) were reported to copulate compared to small or larger groups, but this difference was not found to be significant. Still, the trend downward may support the idea that fewer potential partners result in less copulatory behavior.

Subjects in research facilities (76.9%) were more likely to copulate than those in zoos (63.1%) or sanctuaries (51.6%). An earlier study found a similar proportion of laboratory-housed chimpanzees seen to copulate (71%) [16], but did not sample zoo or sanctuary populations. One of the sanctuaries we surveyed has a relatively large proportion of chimpanzees who have come from private ownership and/or entertainment industry locations, where chimpanzees are much less likely to have experienced adequate socialization with conspecifics; this would be expected to have a negative effect on copulatory behavior [16]. In addition, as we did find that older adult or elderly MR chimpanzees were less likely to copulate than younger MR, it might be expected that zoos and sanctuaries, where a larger proportion (24% for zoos, 15% for sanctuaries) of the sample fell into the elderly category, would have a lower percentage of animals copulating than in research facilities, where only 11% were elderly. In addition, one factor we did not query, but which may be relevant, is the use of oral contraceptives for female chimpanzees, which has been shown to reduce swellings and, in some cases, may reduce or eliminate sexual behavior [56] (though this is not a consistent finding [57]). Zoos are the only facility type assessed here which intentionally breed chimpanzees so they may be generally less likely to use any method of birth prevention, but even in zoos breeding is not always recommended, and when it is not, oral contraceptives are commonly used. In sanctuaries, vasectomies have historically been used as birth control, which would not be likely to affect sexual behavior [58], but due to failed vasectomy procedures, oral contraceptives are now used in addition to vasectomy procedures [59] in at least one sanctuary surveyed. Research facilities also use oral contraceptives, but may often use other strategies for birth control, such as same-sex groupings. For this survey, while there were no zoo-living subjects excluded due to lack of access to the opposite sex, there were 24 excluded from sanctuaries and 218 from research facilities. Our exclusion of chimpanzees’ copulatory behavior for those housed in same-sex groups could incidentally remove animals who do not copulate for other reasons, such as rearing history, from the sample, biasing results toward research facilities. Further investigation is needed to determine what drives the facility-based difference in copulation reported here.

Copulatory behavior is likely impacted more strongly by early rearing history [16,25] than the other STBs we assessed and therefore may be more a reflection of prior experiences than an indicator of current welfare. Based on our findings, rearing history had at least as big an impact on copulation as current factors, such as group-size, and that impact was nearly doubled again for NOTMR chimpanzees who were also male (Table 7).

These data on the prevalence of copulation behavior are important not only because copulation is a behavior related to an animal’s individual fitness, but also because it has been considered an indicator of positive welfare [7,38]. However, Cronin and Ross [23] examined whether there were differences in the expression of welfare-related (atypical) behavior between groups that were permitted to breed and rear offspring, compared to those for which breeding was limited. They found no such differences between these groups and suggested that the relationship between natural reproductive behavior and psychological wellbeing is yet to be completely validated. The expression of copulatory behavior and its relationship with animal welfare may be distinct from the effects (or lack thereof) of actual reproduction and rearing, as assessed by Cronin and Ross [23], but further research on this topic is warranted. Hopefully, the data presented here can provide a starting point for better understanding the relationship between copulatory behavior and welfare.

### 4.2. Tool-Use

Over 90% of the chimpanzees in this study were observed to have used tools. This high prevalence may be because there are sufficient opportunities to observe and learn tool-use in most captive conditions, including but not limited to the provision of tool-mediated enrichment that motivates chimpanzees to use tools to retrieve food. Nonetheless, several of our predictor variables influenced the prevalence of this behavior. Immature subjects were less likely to use tools compared to adults, zoo-living subjects were less likely compared to research facility subjects, and NOTMR males were less likely compared to MR females (Table 7). Tool-use is likely a socially-learned behavior [39,42], typically developed by age six for wild chimpanzees [60]. This likely explains why immature chimpanzee subjects in this study (about a third of whom were less than 6 years old) were less likely to use tools than mature subjects who may have had the opportunity to observe and practice such behaviors.

Looking at the proportions of each subcategory of sex and rearing (Table 3), it is apparent that while males and females who were MR and females who were NOTMR were almost always (upwards of 95%) reported to use tools, male NOTMR were slightly less likely (91%). The magnitude of the difference between males and females who are NOTMR is small, but the sex-by-rearing interaction was predictive: NOTMR males were less than a third as likely as MR females to use tools. Here, the impact of rearing seems to be exclusively on males, and while there is a sex-based difference based on observed, sex-based differences in social learning [36,37,60,61], it is less clear why non-mother-reared males would be affected differently from non-mother-reared females.

Chimpanzees at zoos were less likely to have been seen using tools compared to research facilities. While cognitive research is growing in zoo environments [62], the history of such work is more robust in research facilities [63,64,65]. Such research may have facilitated greater exposure to a range of tools, and devices that require the use of tools. However, as many chimpanzees have been relocated to zoos or sanctuaries in recent years, and researchers interested in chimpanzee cognition have relocated [66], future results may be altered. As our data were rather simplistic, and did not ask about type of tools, tasks, or goals for tool-use, it seems it would be important to explore those variables in further understanding how tool-use might represent behavioral flexibility and signify an animal’s welfare.

### 4.3. Nest-Building

We found that 58.9% of our sample had been observed making nests. This represents a significant deficit. Nest-building, like tool-use, is a behavior learned by chimpanzees in the wild at a young age [39]. Bernstein [42] suggested young chimpanzees learn to build nests by the age of two, based on his observations of wild-born, captive living chimpanzees, and Goodall [39] described wild chimpanzees observing and learning from their mothers before starting to build their own nests at age three. Given that 92% of our sample of immature chimpanzees were at least four years old, we would expect at least 90% of immature subjects to build nests, but only 29.5% were reported to do so.

Elderly subjects were significantly more likely to build nests than adults or immatures (see Table 4). While collinearity tests indicated that rearing and age category could be included together in regression analyses, the elderly group is biased toward mother-reared, predominantly wild-born subjects (94 of 155 elderly chimpanzees), and only 6 WB individuals were not also elderly (Table A1, Appendix B). Considering that wild-caught chimpanzees younger than one year old may be less likely to survive the transition to captivity, it is possible that wild-born, captive chimpanzees would have already learned to build nests at the time of their capture. In our sample, 89.5% of the wild-born chimpanzees were reported to build nests. Mother-reared but not wild-born subjects were significantly less likely, at 72.3% (Table 4). Indeed, analyses of wild versus captive-born, mother-reared subjects indicate that it is the wild-born versus captive-born dichotomy that predicts nest-building, not age. These findings are consistent with prior research [41,42,67], and support posited theories that there is a sensitive period for learning nest-building which is experienced by wild-born subjects before their removal from the wild which provided appropriate exposure to nest-building [67].

Lack of appropriate materials with which to demonstrate nest-building and/or lack of exposure to nest-building in nursery environments could negatively impact development of this skill, perhaps disrupting generation-to-generation transmission. Based on their wild counterparts’ behavior, MR individuals would be predicted to build nests at near 100%, but only 75.5% were reported to do so. Only 45.7% of NOTMR subjects were reported to build nests, but this result is less surprising given that nursery rearing conditions may not have included attempts to demonstrate or teach nest-building skills, particularly if there is indeed a developmental window which facilitates learning this skill during early years.

Zoo-living subjects (92.2%) and sanctuary subjects (55.3%) were more likely to build nests than subjects living in research facilities (48.1%). With all other factors accounted for, living in a zoo increased the likelihood of a subject building nests by 14 times, and doubled the likelihood for subjects living in sanctuaries (as compared to research facility subjects). Since the population of chimpanzees in sanctuaries is primarily made up of animals relocated from research facilities, it seems there is a deficit in research environments that may still be, at least partly, ameliorated, such that some older chimpanzees learn this skill once they move to sanctuary (although note that the increase from 48.1% to 55.3% nest builders is not a large magnitude of difference). Historically, research facilities may not have provided enough nesting materials to stimulate this behavior and/or ensure it was modeled for young chimpanzees and it is possible that provision of nesting material is still inadequate in some facilities. While facilities which did not provide any nesting material would have been automatically eliminated from analysis, we did not ask how much or what sort of material was provided, nor did we ask if there were other locations to sleep or rest which did not require nesting (such as hammocks).

Based on studies such as Baker’s 1997 research [68], which reported a reduction in atypical behavior after adding straw and forage material to chimpanzee enclosures, the provision of nesting materials is currently a standard part of behavioral management programs for captive chimpanzees in all facility-types. Over time, this may increase nest-building, particularly for research facilities. It would be advantageous to assess the use of different materials, in different amounts, for building nests in the interest of providing optimal materials (in kind and in volume) which would promote and support nest-building behavior in captive chimpanzees. Similarly, it may be helpful to provide nesting materials and demonstrate nesting behavior in nursery environments. While most facilities are focused on reducing time any chimpanzee infant is in nursery care, and programs to foster infant chimpanzees or otherwise reintegrate chimpanzees at very young ages are increasingly successful, it may be worthwhile for human caretakers to demonstrate nest-building regularly until a young chimpanzee has been successfully integrated with conspecifics.

### 4.4. Social Grooming

About 85% of our sample engaged in active social grooming. Active grooming of a social partner is a critical component of chimpanzee social behavior [44,45,47,69]. It is, then, a concern for an estimated 15% of our sample to be lacking in this behavior. Immature subjects were 5.3 times more likely than adults to engage in active social grooming. Our results contradict previous research which found that captive chimpanzee adults groomed more than younger animals [70]. Active grooming has been observed, in wild female chimpanzees, to develop by age two, but this grooming does not extend beyond the groomer’s mother until around age five [71]. As our immature category included all subjects up to age 11, it would be expected, based on previous research, that nearly all animals in this group engaged in active grooming behavior, and nearly all of them did so. It seems that some factor other than age has resulted in a deficit in active grooming for adult and elderly subjects.

Females in our sample were about twice as likely to engage in active social grooming as males. Similarly, captive-living female chimpanzees have been reported to groom more than males, at least at younger ages [70]. Since chimpanzee males favor grooming their mothers, other males [72,73] or females in estrus [69,70], it could be that removal from mothers, impact of birth control on estrus, and/or fewer available males to serve as grooming partners are the causes of the sex difference we found in our sample. We cannot determine the potential impact of either birth control or social group sex demographics as we did not ask about either in this survey. However, we can confirm that rearing history had a significant impact: MR chimpanzees in our sample were about five times more likely than NOTMR to engage in active social grooming (Table 7).

In our sample, animals living in zoos were three times more likely than those in research facilities to engage in active social grooming (Table 7). It is possible that the structure of social groups in zoos facilitates more grooming, particularly for male chimpanzees, as increasingly more zoos house large, multimale groups. In the past, zoos often moved adolescent males out of their natal groups due to risk of incestual breeding and/or perceived increases in agonism as males matured. As research has borne out the benefits and lower-than-expected costs (in terms of aggression) to maintaining males in their natal groups [74,75] as would follow the natural dispersal pattern for chimpanzees [43], the composition of zoo chimpanzee social groups is changing. The higher proportion of zoo-living chimpanzees reported to engage in active social grooming may reflect the impact of this management change, particularly for males, as they may now have more available, preferred targets for grooming behavior (more males and more available mothers).

Research facilities, caring for their retired-in-place population of chimpanzees, and sanctuaries, where increasingly more retired research chimpanzees are housed, may work toward larger, multimale social groups as well. This effort is complicated because while some accidental births still occur, research facilities and sanctuaries do not intentionally breed. Thus, the ratio of MR to NOTMR individuals in these facilities’ relatively static populations is not likely to change, and if age does impact social grooming, aging populations of animals may be less likely to groom. Increasing the diversity and male: female sex ratio of chimpanzees’ social groups may be the best way to improve the number of animals who engage in active grooming, however, particularly for facilities where breeding does not occur.

### 4.5. STBs, Mother-Rearing, and Male Chimpanzees

We found that male chimpanzees, as compared to females, were significantly less likely to copulate, use tools, build nests, engage in active social grooming, or perform all four STBs (Table 2, Table 3, Table 4, Table 5 and Table 6). There are sex differences reported for wild and captive-living chimpanzees in age of development and/or frequency of some STBs. Wild adult males are reported to spend less time building nests and to develop nest-building skills more slowly [76] and also have been reported to develop socially-learned and prosocial behaviors more slowly than females [71,72]. A similar sex difference in age of competency has been reported for captive-living chimpanzees [70]. However, these and other studies confirm that males do actively build nests, use tools, copulate, and engage in active social grooming, even if they do so at a later age, to a lesser degree, or with fewer or different partners in comparison to females [43,69,70,71,72]. It does not seem reasonable, based on these studies, that we would see a consistent sex-based difference in the STBs we assessed here. Indeed, regression analyses indicated that sex alone was not a significant predictor of any STB other than active grooming. Copulation, tool-use, and all four STBs were significantly impacted by a sex-by-rearing interaction in which NOTMR males were significantly disadvantaged as compared to MR males and NOTMR females (Appendix C).

Our definition of rearing was rather basic, such that any chimpanzee known to have spent 6 or more of the first 12 months of life with a conspecific mother was categorized as mother-reared, as were all wild-born chimpanzees. Still, we found significant differences in STBs based on these rearing categories: regression analyses indicated that NOTMR chimpanzees were negatively impacted by their rearing history as per all STBs we assessed other than tool-use. Thus, while being male reduced likelihood of performing STBs only in conjunction with being NOTMR (other than as per grooming), being NOTMR had significant impact on its own, as well as impacting some STBs further for NOTMR males. It is well established that rearing history can significantly impact adult chimpanzee physiology [77,78] and behavior [25,55,77,79]. For example, one study showed that chimpanzees who were not reared by their conspecific mother have, as adults, higher hair cortisol levels than those who are mother-reared [78]. Nursery-reared chimpanzees have also been shown to exhibit, as adults, significant and negative differences in brain morphology as compared to mother-reared chimpanzees [77]. These observed differences could be related to a number of factors, including composition of milk formula (as fed in the nursery) as compared to the composition of breast milk [77]. Factors related to deficits in STBs for males appear to be compounded by nursery-rearing in a way that is not clearly understood. Further research to better understand these issues is strongly recommended.

### 4.6. Intervention

A survey of the literature presents limited information with which to compare our STB data. For example, because we focused on simple yes/no outcomes, it would be challenging to compare our data to quantitative data. Instead of reporting who did or did not groom, studies have reported how much, or whom, individuals groomed. Still, if we compare the information gathered here to previously reported research on captive and/or wild chimpanzees, we can clearly see that there are deficits in the currently assessed population and identify not only particular behaviors which may require intervention, but also which chimpanzees might benefit the most from such efforts (Appendix C: Figure A1, Figure A2, Figure A3, Figure A4, Figure A5, Figure A6 and Figure A7).

Nest-building and copulation are two such behaviors which clearly require some intervention. If it is accurate to say, as some researchers have posited, that nest-building is best learned during an early developmental window [42,67,80], we may have missed an opportunity for chimpanzees currently over the age of four. However, focusing on ensuring adequate provision of appropriate nesting materials (and determining which and how much materials to provide) to all captive chimpanzees should have a significant impact on this behavior over time, particularly if the wild-born individuals still in the population demonstrate for younger animals. This suggestion is supported by our findings that sanctuary-living chimpanzees, most of whom relocated from research facilities, are more likely to build nests than their counterparts who have not relocated. In addition, as it seems NOTMR individuals are at a disadvantage, ensuring that caretakers in nurseries model nest-building and provide adequate and appropriate nesting materials to nursery-reared chimpanzees is indicated. While this may not change the conditions which lead to a chimpanzee failing to build a nest, it would, hopefully, allow the ‘taught’ chimpanzee to model the behavior for others in his/her social group, which could assist in maintaining the behavior in captive chimpanzee culture for future generations.

Our findings regarding copulation are a bit trickier to interpret. There seems to be a clear effect of sex and rearing, one which disadvantages NOTMR males, though there appear to be benefits for animals in research facilities and for animals in groups of eight or more chimpanzees (Appendix C: Figure A1, Figure A2, Figure A3, Figure A4, Figure A5, Figure A6 and Figure A7). The potential impact of these and other factors, such as method of birth control, need to be explored. However, at minimum our results support increased effort to reduce nursery rearing and maximize exposure of very young chimpanzees to their conspecifics.

Tool-use is an important STB, as positive welfare assessments focus increasingly on aspects of behavioral flexibility in zoo-housed animals [8,9] as well as in humans [81] and wild animals [82]. The ability to adapt behaviorally to changing environmental features is critical for species survival [83] and could be considered a hallmark of primate species’ survival ability. So, while it is favorable to find that over 94% of the population surveyed exhibited the flexible, adaptive behavior of tool-use, it would still be appropriate to dedicate energy toward improving these numbers. While it is not clear that we should expect 100% of all chimpanzees over the age of 3 to use tools, it is relevant that we had quite a low threshold for considering an animal to be a tool-user: at least once in the past two years. This makes the proportion of subjects who did not use tools seem more problematic. Perhaps increasing motivation would be relevant; we did not ask about the frequency or type of tool-use opportunities which were present for the chimpanzees, and this would very likely be highly relevant.

Based on findings regarding the effect of models [84] and the influence of ‘need’ based learning [85], we should be able to ensure that the remaining 5–6% of the captive chimpanzee population develop this behavior at the expected age. Research has shown that model type affects chimpanzee learning, i.e., live conspecific versus human or videotaped conspecific [84], and that close kin-based and social-relationship-based connections impact the transmission of novel tool-use behaviors [86]. Selection of the most effective models may assist in ‘teaching’ chimpanzees to use tools, perhaps choosing human models for chimpanzees who are human-oriented.

Active, social grooming may be best addressed by continuing efforts to maintain chimpanzee groups in higher male: female sex ratios, with less dispersal of natal males and more dispersal of natal females. Larger social groups will also increase active grooming. It may be worth investigating the impact of larger, more complex enclosures on grooming, as well as habitats which encourage more fission–fusion social groupings since they may result in higher rates of social behavior when individuals reunite [87]. Investigation into these possibilities could be beneficial toward increasing grooming rates overall in captive chimpanzee groups.

Maintaining males in their natal groups, allowing more long-term male associations to develop, and focusing on creating larger social groups in which females are relocated as they enter puberty, at which point they may be approved to breed and thus do not require birth control, may go a long way toward improving STB behavior, particularly for males. It seems clear that particular attention needs to be paid to male chimpanzees, especially if they are not mother-reared. It would be difficult to figure out a way to address problems that may result from the presence of human caretakers in chimpanzee nursery environments, which can lead to stronger orientation towards humans [55,79,88]. It is likely that human orientation would be higher for NOTMR chimpanzees with more human experience in younger years (e.g., reared in a human home versus nursery-reared with conspecific peers), a factor which we did not query by survey, but which has been shown to impact adult chimpanzee behavior [55,88], even within the broader category of nursery-reared individuals [79]. While in some cases stronger human orientation has been associated with positive outcomes [89,90,91], in others, there have been negative outcomes [55,79,88]. In any case, it is reasonable to suggest that socially-learned behaviors may be impacted by increased orientation towards humans, rather than conspecifics, for example, such that NOTMR chimpanzees may not benefit as much as their MR peers from conspecific models of tool-use, nest-building, or other behaviors typically learned by observation [60]. Based on the data we have reported here, it is of greater importance for males to be reared by conspecific mothers, such that they should perhaps be prioritized for fostering to any available surrogate.

We also wanted to identify sources of potential weakness based on facility-type so that there are clear areas where we might focus efforts on improving the welfare of captive-living chimpanzees, with the understanding that each facility-type may face unique challenges in doing so. For example, our data suggest that we may improve the behavioral repertoire of captive-living chimpanzees by building and maintaining larger multi-male, multi-female groups in which more individuals are mother-reared and the dispersal pattern of females, not males, more clearly echoes that of wild chimpanzees.

However, building larger groups can be challenging for any facility, particularly if the facility can only do so by importing animals from other facilities. Relocating chimpanzees is no simple matter and can have transitory negative effects, such as increased stress levels, as indicated by increased hair cortisol [92], and reduced immunological capacity [93]. In addition, regardless of relocation, socializations are often stressful; in wild chimpanzees, the immigration of new females is associated with increased urinary cortisol [94], and in captivity, socializations, particularly those involving multiple males, have been associated with increased wounding [95]. Certainly, this should not preclude the transfer or socialization of most healthy individuals, particularly when enclosure design supports a potentially slow, multi-step integration of new members into an existing group (i.e., [96]. Some facilities may not have adequate enclosures, however, and may not have the finances or space to build them.

These challenges are heightened for research facilities and sanctuaries where populations of chimpanzees decrease due to relocation, natural attrition, and lack of breeding. Population demographics in these facilities will increasingly skew toward elderly individuals, and there is not an option to increase the proportion of chimpanzees who are mother-reared. It will be of increasing importance for research facilities and sanctuaries (and perhaps zoos) to work together to determine which chimpanzees might benefit from relocation toward the goal of creating and/or maintaining larger social groups. Depending on the conditions in which an individual chimpanzee is currently living, it may still be in his/her best interest to relocate or join a new social group, even if the animal is not in perfect health. It would be helpful to further investigate the impact of transfer and/or inclusion in a larger social group on the health and stress level of aging chimpanzees as well as on the behavioral repertoire of chimpanzees who may have been, at the time of this survey, deficient in performance of the assessed STBs. This would contribute toward better management of aging populations and each individual chimpanzee’s maximum welfare.

Useful dialogue comparing animal management procedures across facility-types may lead to a better understanding of how some of these behavioral differences have arisen and may point the way toward therapeutic interventions to promote more STBs, including assessment of how much of any behavior is ideal and the degree to which any behavior, atypical or species-typical, accurately reflects an animal’s welfare. Additional methods developed for assessing chimpanzee welfare in a variety of types of facility have been used to evaluate some sanctuaries [97] and may be of use for comparison, evaluation of, and improvement of any facility, regardless of its classification as zoo, research institution, or sanctuary, which houses chimpanzees.

## 5. Conclusions

While it is typical to collect information regarding the atypical behavior exhibited by animals in captive conditions, it is less common to assess the species-typical behaviors which those individuals do or do not exhibit. Here, we have used a rather broad, simplified survey to gain a better picture of the degree to which a very large sample of chimpanzees living under human care in the United States are exhibiting four behaviors considered to be typical of their species in their natural habitat: tool-use, social grooming, nest-building, and copulation. In addition, the ability to characterize 100 WB individuals’ expression of four STBs after decades living in captivity was unique and valuable, as these individuals are older and reaching the end of their lifespans. Though these individuals are aging out of the captive population, it was worth analyzing and reporting the data we collected on these WB chimpanzees as it may help us understand more about the etiology of various behaviors.

It is important to keep in mind that surveys can be biased more easily than some other types of data, as results can be affected, for example, by the method of response collection [98]. For this study, we asked for basic data from several facilities, many of which were bound to differ in the amount of time individual chimpanzees were observed, the method of recording or collecting behavioral information, and the experience level of people who are observing the chimpanzees. By focusing on easily identifiable behaviors and very basic information about those behaviors we hoped to reduce some of the inconsistencies which may have resulted if we had asked, for example, for frequencies of behavior or information about more easily misidentified behavior. However, the use of a simple survey to collect information about a large sample of chimpanzees did allow us to obtain a better, wider view of the state of the captive chimpanzee population in the United States than would have been possible with more involved methods of data collection. The survey reported here should be considered as a starting point for understanding what sort of STBs are observed in captive chimpanzee populations in the United States and which of those STBs may be most notably deficient, as well as providing us some starting points for further evaluation and for potential ‘treatment’ of deficits. It is possible that a survey method such as we employed here will allow us not just a starting point for improving the welfare of chimpanzees in captivity, but also allow us to assess changes over time in the behavioral diversity profile of United States chimpanzees.

## Figures and Tables

**Table 1 animals-13-00251-t001:** Forward Wald regression statistics for best models and significant predictors.

STB	Stat	Age = Immature	Age = Elderly	Sex = Male	Rearing = NOTMR	Facility-Type = Zoo	Facility-Type = Sanctuary	Group-Size = Pair	Group-Size = Bigger	Rearing by Sex = NOTMR, Male
Copulate (*n* = 788)	B	--	--	--	−0.77	−1.24	−1.30	−0.53	0.755	−0.97
	SE	--	--	--	0.21	0.22	0.24	0.32	0.20	0.23
	Wald	--	--	--	13.89	30.81	30.42	2.71	14.18	17.84
	*p*	--	--	--	**<0.001**	**<0.001**	**<0.001**	0.100	**<0.001**	**<0.001**
Tool-use (*n* = 880)	B	−2.14	−0.59	--	--	−1.07	1.10	--	--	−1.31
	SE	0.36	0.54	--	--	0.36	0.75	--	--	0.36
	Wald	35.17	1.16	--	--	9.02	2.13	--	--	35.17
	*p*	**<0.001**	0.281	--	--	**<0.005**	0.144	--	--	**<0.001**
Nest-building (*n* = 880)	B	--	--	--	−1.32	2.62	0.703	--	--	--
	SE	--	--	--	0.16	0.31	0.20	--	--	--
	Wald	--	--	--	67.94	73.13	12.96	--	--	--
	*p*	--	--	--	**<0.001**	**<0.001**	**<0.001**	--	--	--
Active Social Groom (*n* = 975)	B	1.67	−0.63	−0.88	−1.74	1.14	0.247	--	--	--
	SE	0.61	0.36	0.21	0.27	0.39	0.27	--	--	--
	Wald	7.60	3.13	16.94	41.67	8.32	0.861	--	--	--
	*p*	**<0.01**	0.077	**<0.001**	**<0.001**	**<0.005**	0.354	--	--	--
All Four STBs (*n* = 788)	B	−1.55	−0.07	--	−1.14	--	--	0.43	1.05	−1.12
	SE	0.33	0.24	--	0.19	--	--	0.31	0.17	0.26
	Wald	22.16	0.080	--	35.86	--	--	1.91	37.92	17.82
	*p*	**<0.001**	0.777	--	**<0.001**	--	--	0.167	**<0.001**	**<0.001**

B = regression equation coefficient; SE = standard error for B; Wald = Wald statistic; *p* = probability/Wald statistic. Reference category for Age = Adult; Sex = Female; Rearing = MR; Facility-type = Research; Group-size = Smaller group; Rearing by Sex = MR/Female. Bolded *p*-values indicate significant predictors retained by the best model; -- = Not retained by best model.

## Data Availability

Data available on request to corresponding author and Emory National Primate Research Center.

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
