# Peer review of "Chimpanzees (Pan troglodytes) in U.S. Zoos, Sanctuaries, and Research Facilities: A Survey-Based Comparison of Species-Typical Behaviors"

_animals, 2023, doi:10.3390/ani13020251_

Round 1

Reviewer 1 Report

Previous research in the domain of chimpanzee welfare has primarily focused on atypical behaviours as an indicator of negative affective states and poor welfare – here the authors focus on species typical behaviours, which have often been overlooked. They underscore the importance of looking at both of these behaviour types when examining captive welfare and propose several interventions based on the deficits seen in the chimpanzees assessed.

This research article is well-written, with a clear understanding of chimpanzee behaviour apparent throughout. It boasts an impressive sample size of over 1000 individuals, across three different facility types, and has the potential to gain a broad overview of the presence of species typical behaviours in captive chimpanzees.

Nevertheless, there are several points that I believe warrant further attention:

1.       I’m a little concerned about the statistics used in this paper, which then also affects the validity of the findings… I have always been taught that stepwise model selection is an unreliable (and, in statistician circles, unscientific) method for constructing a model. I’m no statistician myself, so I am happy to be proven wrong – but it would certainly be useful to include a few sentences to explain why you chose this method (I know there are also a few different model selection methods around)/why it is appropriate here. If there is no clear support for this method over alternatives (e.g. logistic GLM with full model) then I would suggest 1) running a logistic GLM or other appropriate model and including that instead, or 2) excluding this analysis and only including the chi-square results (as you say – this allows the inclusion of a larger sample anyway). If you do include both a regression analysis and chi-square tests then please be clear throughout which results you are referring to/which one is most relevant, as this is currently somewhat confusing.

Also related to the Wald test – I would be very surprised if it is not possible to change the reference category (L229). Hopefully you could find someone to show you how to do this, as it would be important to include all comparisons in the results.

Finally, please provide more details about which program was used for the statistical analyses.

2.       There are several points where it is hard to follow the story of the paper. For example in the introduction you mention that studies have used STBs as markers of positive welfare, but then go on to say that it can be difficult to ascertain which behaviours really do indicate positive welfare, and that similarities between captivity and the wild don’t necessarily indicate similar welfare states, but then go on to say that behaviours that occur frequently in wild chimpanzees may be linked to positive welfare and you will focus on (some of) these. The aim of the study is outlined to be to characterise prevalence of STBs, investigate factors affecting their expression and promote environments that support increased expressions of these behaviours to improve welfare. At this point the rationale is hard to grasp, as it’s been explained that it’s not clear which STBs are truly markers of positive welfare, and even if these are identified, it is unknown to what extent the behaviours should be increased in order to achieve optimal welfare (e.g. is the aim for 100% of individuals to show these behaviours in X amount of time? For all ages, sexes and facilities to show the same % of behaviours? Or to show them at similar levels to the wild? But it was noted that similarities between captive and wild chimps do not necessarily indicate similar welfare states. What is the justification for choosing an arbitrary aim?)

Similarly, in the discussion there are instances where the findings are outlined, but then simply ‘explained away’ by a limitation in the study or by some previous research – leaving the reader with very little concrete, novel, information to take home. Try to keep a coherent and logical story, and focus on the most central points and take home messages, rather than discussing and explaining each finding.

3.       In terms of the STBs that were selected here – please add a few sentences to explain why social play was not included (in the introduction it is noted that social play is an important STB to include in future studies). Equally, if the choice was based on being behaviours that occur frequently in wild chimpanzees (as these may be linked to positive welfare), then why not also include aggression?

Additional points to address:

Line 85-91: Would some of these arguments and limitations also apply when looking at a presence/absence of species typical behaviours? For example, if nest-building is not learned early on (perhaps when growing up in a very restricted research setting), then this might also not be observed once the individual is moved to an improved environment? It would be good to elaborate here on how species typical behaviours might be a more reliable indicator of current welfare, than atypical behaviours.

Line 119-121: “STBs queried were chosen as they are arguably related to welfare…” – it’s not entirely clear how e.g. tool-use and nest-building, are necessarily related to welfare (if there are no situations that require tools, why would an individual need them? Why build a nest if there is already a comfortable sleeping spot?). Please add a sentence or two to explain this when you propose these.

Line 124-125: again, make it clear how tool-use being a complex ability and a marker of behavioural flexibility makes it interesting for this study

Line 134: clarify whether you mean initiation of a social grooming bout, or simply actively grooming the other. Be consistent with this throughout.

Line 162-163: add a brief sentence on why you chose two years (why not one? Do you expect this would have given very different results?)

Line 257: Figure 1. repeats what is already said in the text – only include one or the other. If keeping figure then also add Ns.

Figures 2-5.: key is unclear – make sure this is visible.

Line 382-474: this section could be significantly cut down. First paragraph is a summary of what we learned in the results, however the second paragraph then explains that it is unclear whether copulation is actually an indicator of positive welfare. This makes it difficult for the reader to place much importance on the subsequently discussed findings - remind the reader why they are still relevant. L413 also suggests that we already know that early rearing history has a significant impact on adult copulatory behaviour, so that it perhaps doesn’t make much sense to use this as a measure of current welfare – then a little confusing as to why it was included here (please clarify). In this section, also make sure that each paragraph is a distinct, clear point (L420-436 paragraphs are seemingly all part of the same point) and ensure that all information introduced is directly linked back to the aim of the paper. Sometimes it feels like there is a lot of interesting information, but a coherent story or argument is missing.

Line 383: did you exclude individuals where the only opposite-sex individuals in the group were kin e.g. sister living with brothers?

Line 414-419: I’m not convinced that it’s very helpful to compare the exact percentages found here, to King et al. They had a much smaller sample of 25 males and 46 females and differed in several other ways – so any variances in findings could well be explained by differences in methodology.

Line 417-419: but should be noted that in the chi-square tests there was a significant difference in sex (also noted on L385-386). It’s a little confusing for the reader to have these two statistics – explain which one is relevant and when.

Line 475-502: again I’m not sure about the importance or relevance of the tool-use STB. From reading this section it seems that all findings were simply in line with previous studies/unsurprising e.g. immatures less likely to use tools = not surprising as have had less time for social learning [34, 37, 52], zoo-living individuals were less likely to use tools than research facility subjects = likely have less exposure to tools and conditions that require tools [55-57], NOTMR males were likely to use tools than MR females = likely due to sex-based differences in social learning [31, 32, 52, 53]. Make sure to always highlight to the reader what the current study is adding to our existing knowledge of chimpanzee behaviour – how is it new, exciting and important? As a reader I am also cautious about placing too much weight on the tool-use findings, without knowing further information about how often potential tools were available, and how often the chimpanzees encountered situations that would call for tool-use.

Line 511-513: similar to above - hard to assess this finding without knowing more about the nesting materials available, and the sleeping substrate etc. at the different facilities. Perhaps individuals have the ability and skill to make nests, but no motivation/need (there are already comfortable places to sleep – so no need to use energy on making a nest), or they have the ability, but little or no suitable nesting materials or spots.

Line 573-574: should be noted that they had very small sample (6 subjects) and very different sampling method (2min scan sampling), also not looking at initiation only – so not an ideal study for comparison. Perhaps replace this with a more appropriate citation, or make it clear to the reader that these differences exist.

Line 707-708: a very high proportion of the individuals in your sample use tools (94%), and not clear how many opportunities they were all given, so the numbers of individuals capable and motivated to use tools it likely even a little higher. Is it really necessary to dedicate energy toward improving these numbers even more? From the current data we don’t know if perhaps all the chimpanzees are doing one repetitive movement with the same tool (not very flexible - and not sure eliciting this behaviour from the remaining 6% would greatly improve their welfare), or perhaps they are all inventing tools in creative new ways. In the wild some chimpanzees virtually never use tools while others are much more explorative and inventive – what is the reasoning behind aiming for all captive individuals to show this behaviour (seems unnatural)?

Line 719-722: a naturalistic social and physical environment should hopefully already be the aim for chimpanzee enclosures, in order to mirror wild conditions where possible, but it doesn’t do any harm to repeat this

Tables 1-4 and Figures 2-8 should go into a supplementary materials file, if it is possible to have one, as they are not essential but still interesting if a reader wants to look into more detail.

Reviewer 2 Report

In the manuscript animals-1866560 the authors present a detailed analysis of a survey to zoos, research facilities, and sanctuaries which housed 1122 chimpanzees to relate the occurrence of four species typical behaviors with the animals’ age, sex, social group size, rearing history, and enclosure type. Based on their results, they provide some recommendations to improve the welfare of captive chimpanzees. Overall, I think their approach of focusing on species typical behaviors to evaluate some of the factors that may influence these behaviors is valuable. Although some of their recommendations may not be easily appliable by the facilities, the authors provide some objective and useful guidelines to improve chimpanzee management.

Besides some minor comments, described below, I have two important suggestions. The first one is to explain in the Methods why a two-years period was selected for the survey. The second one is to consider in the Discussion the low sample size of the sanctuaries when analyzing the effect of facilities on the occurrence of behaviors.

In the lines below, I present my comments and suggestions in more detail.

Methods

L154-156 I suggest the authors to present the criteria they used to select the facilities that participated in the survey.

L175-178 It seems that group size did not change in the 2-year period when the occurrence of the behaviors was reported, is this correct? If there was a change in group size in a facility, was it possible to identify if there was also a change in the occurrence of a behavior?

L188-190 The references in the text of Tables 5 and 7, with no reference to Table 6 is confusing, especially because both tables (5 and 7) appear just below that paragraph. I suggest the authors to rephrase this sentence to make it clear that these tables are part of the Results and to present the tables in the correct order.

Discussion

L631 Figure 7 is mentioned but there were no previous mentions of Figures 2 to 6 (references for these figures are only in L660).

L 756-757 I suggest the authors to consider here the fact that their sample size of sanctuaries was only of two.

Tables and Figures

Table 1 Please, explain in the legend the meaning of the acronyms.

Figures 2 to 7 Please, correct the errors in the legends of the figures to include all the behaviors.

Author Response

Thank you for your comments and advice for improving this paper. My responses will be in italics below:

Methods

L154-156 I suggest the authors to present the criteria they used to select the facilities that participated in the survey. (Added more about this)

L175-178 It seems that group size did not change in the 2-year period when the occurrence of the behaviors was reported, is this correct? If there was a change in group size in a facility, was it possible to identify if there was also a change in the occurrence of a behavior? (Included information about this in the methods section, including reasons for not including if changes in group size occurred)

L188-190 The references in the text of Tables 5 and 7, with no reference to Table 6 is confusing, especially because both tables (5 and 7) appear just below that paragraph. I suggest the authors to rephrase this sentence to make it clear that these tables are part of the Results and to present the tables in the correct order. (put in different places and changed reference locations to tables as well)

Discussion

L631 Figure 7 is mentioned but there were no previous mentions of Figures 2 to 6 (references for these figures are only in L660). (moved to supplementary text)

L 756-757 I suggest the authors to consider here the fact that their sample size of sanctuaries was only of two. (this info is in the supplementary tables - the number of chimps in the sanctuaries was still large, about as large as the number in zoos, despite that there were only two locations)

Tables and Figures

Table 1 Please, explain in the legend the meaning of the acronyms. (fixed)

Figures 2 to 7 Please, correct the errors in the legends of the figures to include all the behaviors. (Fixed)

Reviewer 3 Report

This study is an important survey of species-typical behaviors (STP) in captive chimpanzees as indicators of welfare and guide to how it might be improved.  I have some suggestions for revisions: 1)The STP data are collapsed for each setting and then compared across settings. I found myself wondering about the degree of variation in STPs within each setting (e.g., was copulation recorded in each of the sanctuary settings; was it more variable in some settings than others). 2) Who reported the STPs for each setting? Were multiple individuals queried in each facility? It is difficult to imagine that a single person would remember this detail about each chimpanzee in each facility. If more than one person at each facility provided the information for each chimpanzee, was any effort made to evaluate the reliability across reporters (observers)? 3) Figures 2-8 are very difficult to interpret because there are 5 shaded patterns but only 3 behaviors listed (uses tools, builds nests, initiates grooming) making it difficult to match them. This may be why it looks like the percentages in the figures don't all match the ones provided in the results section: For example,  percent grooming in the results is 91 and 79 percent for females and males, respectively, but figure 3 it looks like it is about 50 vs. less than 40%, assuming the open bars represent grooming percentages. Also, why is copulation not shown in these figures?

Lastly, a more minor point: In several places of the manuscript, the authors state that behaviors promote "species survival" (e.g., copulation). This is a bit antiquated evolutionary language. A more accurate view is that behavior is linked to survival and reproduction (fitness) at the level of the individual. In other words benefits of behavior are to the individual, not the species.

Author Response

Thank you for your time and your comments! I will respond below in italics:

1)The STP data are collapsed for each setting and then compared across settings. I found myself wondering about the degree of variation in STPs within each setting (e.g., was copulation recorded in each of the sanctuary settings; was it more variable in some settings than others). (I think this would be really interesting to discuss but I am not sure I have the space to do it for this paper!)

2) Who reported the STPs for each setting? Were multiple individuals queried in each facility? It is difficult to imagine that a single person would remember this detail about each chimpanzee in each facility. If more than one person at each facility provided the information for each chimpanzee, was any effort made to evaluate the reliability across reporters (observers)? (responded by adding some clarifying text in the methods section)

3) Figures 2-8 are very difficult to interpret because there are 5 shaded patterns but only 3 behaviors listed (uses tools, builds nests, initiates grooming) making it difficult to match them. This may be why it looks like the percentages in the figures don't all match the ones provided in the results section: For example,  percent grooming in the results is 91 and 79 percent for females and males, respectively, but figure 3 it looks like it is about 50 vs. less than 40%, assuming the open bars represent grooming percentages. Also, why is copulation not shown in these figures? (there was a misprint in the figure legends - fixed and moved the figures to supplementary section)

Lastly, a more minor point: In several places of the manuscript, the authors state that behaviors promote "species survival" (e.g., copulation). This is a bit antiquated evolutionary language. A more accurate view is that behavior is linked to survival and reproduction (fitness) at the level of the individual. In other words benefits of behavior are to the individual, not the species. (I addressed this in the text)

Author Response

Response to reviewer 4:

1) I worked on reorganizing the introduction and conclusion to address these concerns and hope that it will read better now.

2) This is a very complicated question but I agree it could be better clarified in the paper - I worked on addressing this issue and tried to keep that from extending the length of the paper

3) I tried to buff up the explanations for choosing these particular behaviors, but it is also true that much of the selection was based on the way we were attempting to gather a quick snapshot of behavior across over a thousand animals so we had to choose based on ease of identifying and observing, as well

4) i used atypical instead of abnormal and edited to be consistent throughout. I also attempted to include a more thorough discussion of welfare without extending the length of the paper.

5) the goal of this survey was intended to be very simple, easy, and quick for all respondents, many of whom had hundreds of animals about whom they had to gather a lot of information; we wanted to make the requirements for an animal exhibiting a particular behavior very low because we wanted someone who didn't spend a lot of time watching the animals to still have some idea if a behavior occurred. Tried to be more clear that this data and analysis is intended to be a starting point, not an ending point, for understanding the relationship between the various behaviors and welfare.

6) provided survey in appendix section; should be self explanatory now!

Minor comments:

1 - addressed

2 - added some info and citations

3 - added info supporting including these animals in some analyses but removed reference to them when it was not relevant/important

4 - edited

5 - edited

6 - edited

7 - added info on how this was addressed

8 - attempted to revise to better explain

9 - tried to make this more clear

10 - thank you!

11 - Made sure this was earlier in the paper

Round 2

Reviewer 1 Report

I am grateful to the authors for addressing the previous comments and I do believe several issues have been clarified and the manuscript has been improved. As the Survey Questionnaire was not previously attached, it is useful to have this now.

However, having looked at this I am hesitant about the validity and reliability of the questionnaire responses. First, the question relating to STBs is simply phrased as “Does the chimp…?”, whereas the following question on abnormal behaviour refers to “In the last two years…” – then presumably the respondents simply tried to remember if the chimpanzee ever engaged in the four STBs, which is at odds with what is reported in the manuscript and this time period likely varies widely between respondents. Second, it seems the respondents were only asked about “tools to acquire food”, which is again not made clear in the manuscript. Third, it is unclear whether respondents were told to skip the questions on tool-use and nest building when there were no materials; L415-416 suggests they were told this, but this is not mentioned on the questionnaire. Perhaps there is a further instruction sheet with more details that was given to the respondents, as the questions themselves are quite simplistic and could potentially be interpreted in different ways.

Overall, while I feel that this paper makes an important contribution given the large sample size and its focus on STBs, the limitations of the methodology make it virtually impossible to draw concrete and practical conclusions. The authors do point out that this paper is merely a starting point, and the findings should be interpreted with caution due to the broad and simplistic nature of the questionnaire - and I completely agree with this – but because of this I think the paper might be better off being severely reduced in length so there is more focus on the method itself and on the broad/general findings, and less emphasis and discussion of each specific finding (especially nest-building and tool-use).

Author Response

The questions raised regarding the survey itself are fair - to be clear, we did specify the two-year period and the necessity to exclude information about certain behaviors if, for example, opportunities to use tools or nesting materials were not provided. We also suggested using tools to procure food as an example, but clarified that this was just one example of tool use. We agree that the survey itself was not specific enough and did require these clarifications. The clarifications were provided by email to all survey respondents but we did not update the actual survey, so I will do that at this time to better represent the questions we asked respondents to answer based on emails. Hopefully that will satisfy issues with the survey itself. We feel that the number of animals represented here are worth reporting, and that the data are worth analyses, though the limitations inherent are important to state, and are stated, in the paper itself. We felt it was important to run exploratory regression analyses to identify potential areas of interest for future research which would not, perhaps, be apparent with fewer subjects (we had data for over a thousand chimpanzees here). Additionally, because hundreds of individuals for whom we had partial data were excluded from regression analyses, we felt it important to run some Chi-square analyses which allowed those individuals to be included. Hopefully we have addressed your concerns at this point.

Reviewer 4 Report

Round 2: 

Overall, many of the original concerns were addressed or explained in a way that was convincing to the reader. I still have overall concerns about the methodology, however, for the purpose of the paper as described in the introduction and discussion, I would say the majority of concerns have moved from “major” to “minor.”

Major Concern: I would still recommend that the authors work to frame the information as exploratory rather than the heavy analytical approach taken so that the real value of the paper can be understood.

Major/Minor Concern # 1: I understand that the authors intend for the data to be a starting point for future studies but I still have concerns about the data collection methodology. Once in a two year period is not a “typical” appropriate time frame. The data collected were sparse in terms of group size changes, origin of the animal, etc. as noted by the authors. These limitations may be more impactful on the results. I understand that the intention is to reduce the labor on the folks answering the survey but the cost of this reduction in labor needs to be weighed with the benefit. I’m not convinced that the data is reliable enough to make the claims that were discussed in the discussion portion of the paper. 

Example: 558: With all of the limitations listed, is this accurate? Not knowing contraceptives, if the opposite sex were related, etc., I’m not convinced by this conclusion.

Major/Minor Concern #2: Atypicality has not been defined - it should not be assumed that people will note it as the opposite of species typical behavior. It is such a key component of the work described, I think this element needs to be crystal clear. 

Mincor Concerns

62: replace or with and

75: Sounds awkward

76: How are health measures different from physiological profiles? Are affective states determined via behavioral expressions? Please consider incorporating “proxies” here as there are redundant/overlapping components in the list of “outputs.” 

126: “wild animals in captive conditions” - do you mean animals that were free-living then captured? If not please, please remove the word “wild” as it has further implications

129 - 132: There are multiple contradictions in these statements - you say that behavioral proxies are strong methods, but then say they are not, but then we should still use them? Please clarify.

135: What is simple behavior? Does that mean there are complex behaviors? I would be cautious in  using that type of language to describe behavior. Try something like, “behaviors that can be operationalized and recognized by folks familiar and unfamiliar with chimpanzee behavior”

158: Avoid using complex vs. simple - maybe try nuanced? Otherwise, good explanation - this definitely strengthens the paper

160 - 162: Used “wide variety” too many times

163-165: Sounds awkward/incomplete sentence

169: “Nevertheless” sounds like you are brushing off the aforementioned limitation in 165-168 - I would use “However” or something a bit softer

169-173: Strong way to conclude - does clarify things for the reader

178: “research facilities in the United States” 

500: This comparison to a previous study muddies your findings, it is not helpful to the reader to have the specific percentages from a previous study. If you had wanted to compare to King et al., why did you use different definitions?

Please check for grammatical errors and typos throughout.

Author Response

Thank you for your constructive comments regarding our paper. I have made edits and highlighted them based on your comments and notes and have hopefully addressed your concerns. The only issue which I did not address is in regards to comparisons to King et al and other previous studies; we felt it important to discuss previous findings along with ours but as the data we collected were different in some aspects we also felt it important to explain how those differences might impact our findings. We did not design the survey in order to facilitate this comparison, however, and made some of our decisions in collecting our data based on definitions that are used presently, but were not perhaps used at the time of the previous studies.

Thank you for your help and your time reviewing!

Overall, many of the original concerns were addressed or explained in a way that was convincing to the reader. I still have overall concerns about the methodology, however, for the purpose of the paper as described in the introduction and discussion, I would say the majority of concerns have moved from “major” to “minor.”

Major Concern: I would still recommend that the authors work to frame the information as exploratory rather than the heavy analytical approach taken so that the real value of the paper can be understood.

Major/Minor Concern # 1: I understand that the authors intend for the data to be a starting point for future studies but I still have concerns about the data collection methodology. Once in a two year period is not a “typical” appropriate time frame. The data collected were sparse in terms of group size changes, origin of the animal, etc. as noted by the authors. These limitations may be more impactful on the results. I understand that the intention is to reduce the labor on the folks answering the survey but the cost of this reduction in labor needs to be weighed with the benefit. I’m not convinced that the data is reliable enough to make the claims that were discussed in the discussion portion of the paper. 

Example: 558: With all of the limitations listed, is this accurate? Not knowing contraceptives, if the opposite sex were related, etc., I’m not convinced by this conclusion.

I reviewed this section and made sure I was qualifying our conclusions/speculation. 

Major/Minor Concern #2: Atypicality has not been defined - it should not be assumed that people will note it as the opposite of species typical behavior. It is such a key component of the work described, I think this element needs to be crystal clear. 

 Atypical and abnormal are two very commonly used terms for describing animal behavior that is not usually seen in the wild or is not seen as frequently etc. Atypical is usually used at present for a number of reasons but this would require an additional paragraph or two of text and that seemed counterproductive at this point. This paper does not focus on atypical behavior, in any case, so it seemed less interesting to elaborate too much on the definition, particularly since it is a very commonly used term.

Mincor Concerns

62: replace or with and

 Done

75: Sounds awkward Adjusted

76: How are health measures different from physiological profiles? Are affective states determined via behavioral expressions? Please consider incorporating “proxies” here as there are redundant/overlapping components in the list of “outputs.” Adjusted.

126: “wild animals in captive conditions” - do you mean animals that were free-living then captured? If not please, please remove the word “wild” as it has further implications Incorporated this edit.

129 - 132: There are multiple contradictions in these statements - you say that behavioral proxies are strong methods, but then say they are not, but then we should still use them? Please clarify.

 This is a complex issue and does not have a simple answer. I have attempted to clarify without reducing the complexity of the issue beyond what is justified. There are contradictions in the research and theories about these behaviors and methods for evaluating welfare etcetera which are beyond the intended scope of this article.

135: What is simple behavior? Does that mean there are complex behaviors? I would be cautious in  using that type of language to describe behavior. Try something like, “behaviors that can be operationalized and recognized by folks familiar and unfamiliar with chimpanzee behavior”

 Incorporated this edit.

158: Avoid using complex vs. simple - maybe try nuanced? Otherwise, good explanation - this definitely strengthens the paper

160 - 162: Used “wide variety” too many times

 Edited.

163-165: Sounds awkward/incomplete sentence

 Edited.

169: “Nevertheless” sounds like you are brushing off the aforementioned limitation in 165-168 - I would use “However” or something a bit softer

Edited

169-173: Strong way to conclude - does clarify things for the reader

178: “research facilities in the United States” 

 Edited. 

500: This comparison to a previous study muddies your findings, it is not helpful to the reader to have the specific percentages from a previous study. If you had wanted to compare to King et al., why did you use different definitions?

 We felt it important to discuss previous findings along with ours but as the data we collected were different in some aspects we also felt it important to explain how those differences might impact our findings. We did not design the survey in order to facilitate this comparison, however, and made some of our decisions in collecting our data based on definitions that are used presently, but were not perhaps used at the time of the previous studies.

Please check for grammatical errors and typos throughout.